

# Object detection in optical imaging of the Internet of Things based on deep learning

Rui Chen[1], Lei Hei[2] and Yi Lai[1]

[1] School of Communications and Information Engineering, Xi'an University of Posts and Telecommunications, Xi'an, Shanxi, China
[2] The Institute of Xi'an Aerospace Solid Propulsion Technology, Xi'an, China

## ABSTRACT

This article endeavors to enhance image recognition technology within the context of the Internet of Things (IoT). A dynamic image target detection training model is established through the convolutional neural network (CNN) algorithm within the framework of deep learning (DL). Three distinct model configurations are proposed: a nine-layer convolution model, a seven-layer convolution model, and a residual module convolution model. Subsequently, the simulation model of CNN image target detection based on optical imaging is constructed, and the simulation experiments are conducted in scenarios of simple and salient environments, complex and salient environments, and intricate micro-environment. By determining the optimal training iterations, comparisons are drawn in terms of precision, accuracy, Intersection Over Union (IoU), and frames per second (FPS) among different model configurations. Finally, an attention mechanism is incorporated within the DL framework, leading to the construction of an attention mechanism CNN target detection model that operates at three difficulty levels: simple, intermediate, and challenging. Through comparative analysis against prevalent target detection algorithms, this article delves into the accuracy and detection efficiency of various models for IoT target detection. Key findings include: (1) The seven-layer CNN model exhibits commendable accuracy and confidence in simple and salient environments, although it encounters certain instances of undetected images, indicating scope for improvement. (2) The residual network model, when employing a loss function comprising both mean square error (MSE) and cross entropy, demonstrates superior performance in complex and salient environments, manifesting high precision, IoU, and accuracy metrics, thereby establishing itself as a robust detection model. (3) Within intricate micro-environments, the residual CNN model, utilizing loss functions of MSE and cross entropy, yields substantial results, with precision, IoU, and FPS values amounting to 0.99, 0.83, and 29.9, respectively. (4) The CNN model enriched with an attention mechanism outperforms other models in IoT target image detection, achieving the highest accuracy rates of 24.86%, 17.8%, and 14.77% in the simple, intermediate, and challenging levels, respectively. Although this model entails slightly longer detection times, its overall detection performance is excellent, augmenting the effectiveness of object detection within IoT. This article strives to enhance image target detection accuracy and speed, bolster the recognition capability of IoT systems, and refine dynamic image target detection within IoT settings. The implications encompass reduced manual recognition

Corresponding author
Rui Chen, chenrui@xupt.edu.cn

costs and the provision of a theoretical foundation for optimizing imaging and image
target detection technologies in the IoT context.

target detection

## INTRODUCTION

In the era of big data, Internet of Things (IoT) technology has emerged as a pivotal driving
force across various domains, signifying the progressive assimilation of IoT into diverse
sectors. This transformative trend has propelled image object detection technology to
the forefront, granting it widespread utility in numerous domains. As IoT continues its
evolution, the growing significance of image object detection becomes increasingly evident.
This technology excels at extracting crucial insights from image data, encompassing object
recognition, detection, and pose estimation. These capabilities are essential to address
the ever-growing requirements of data acquisition, underscoring the pivotal role of
image object detection in propelling IoT system development. The confluence of big data
and the pervasive adoption of IoT technology facilitates the collection and processing
of unparalleled volumes and types of data. This data is sourced from sensors, devices,
and applications and encompasses various forms of image data. Image object detection
technology excels in identifying and tracking objects and proficiently estimates their
positions and orientations. It finds applications in a multitude of domains such as smart
cities, intelligent transportation, agriculture, healthcare, and more. In the realm of IoT
system development, the application domains of image object detection technology
continue to expand. This includes its utilization in monitoring and security systems to
facilitate the identification of potential hazards or anomalies. In the retail sector, image
object detection technology plays a pivotal role in recognizing and tracking goods, thereby
enhancing inventory management. In agriculture, it aids in monitoring plant growth and
health. In healthcare, it finds applications in the domain of medical image analysis and
diagnosis. In industrial production, it contributes to quality control and optimization
of production processes. These illustrations represent only a fraction of the potential
applications as the capacities of image object detection technology remain subjects of
continual exploration and expansion. Efficient image object detection hinges upon the
incorporation of artificial intelligence (AI) technology and the adept utilization of machine
learning methods. The integration of AI technology is indispensable for augmenting image
recognition capabilities, refining recognition techniques, and enhancing the adaptability
of image object detection algorithms (*Tran-Dang et al., 2022*). The realization of AI
technology is contingent on the effective application of machine learning methods.
Deep learning (DL), as an emerging field within machine learning, centers its focus on
comprehending the underlying patterns within sample data and the creation of multilayer
representations (*Shorten, Khoshgoftaar & Furht, 2021*).

DL has gained widespread adoption and serves various practical purposes across different domains. This method proves to be a powerful tool in microscope analysis, surpassing conventional image processing techniques and yielding optimization effects in the investigation of various biological processes (*Von Chamier et al., 2021*). DL algorithms find application in industrial production and construction, enhancing target recognition systems to improve speed and accuracy, thereby boosting construction safety and preventing violations (*Huang et al., 2021a*). The integration of DL-based color and depth image information fusion elevates the success rate of mechanical arm grasping position detection models, attaining higher monitoring accuracy (*Jiang et al., 2021*). DL algorithms effectively address issues such as synthetic aperture radar (SAR) automatic target recognition, even with limited training samples (*Wang et al., 2021*). In oceanic contexts, DL's aptitude for large data feature learning contributes to the creation of ocean image target datasets, bolstering detection accuracy (*Fu, Song & Wang, 2021*). When applied to IoT image target detection, DL approaches may improve feature extraction accuracy through principal component analysis and advancing image recognition quality (*Jacob & Darney, 2021*). DL encompasses diverse single-model artificial neural network structures, including feedforward neural networks, convolutional neural networks (CNNs), recurrent neural networks, and generative adversarial networks. Of these, CNN, frequently utilized for image target detection and effectively classifies large datasets, enhances image quality, and improves recognition outcomes (*Girsang, 2021*). CNN notably advances image target recognition system accuracy, speed, and overall performance (*Qing et al., 2021*; *Bai et al., 2022*). In the domain of radar image target recognition, DL's deep convolution networks enhance SAR image features, thereby elevating accuracy and application potential (*Gao et al., 2019*). Notably, CNN-based image detection and recognition within the DL framework significantly elevates detection efficacy in several sectors, including radar image target recognition (*Huang et al., 2021b*; *Zhang et al., 2021*). Given the transformative impact of DL, integrating CNN into image target detection algorithms becomes imperative. This strategic fusion represents a pivotal step in enhancing image target recognition technology, amplifying computer vision capabilities, and propelling the evolution of IoT systems.

This article presents an innovative approach to bolstering image target detection capabilities within IoT systems through the integration of CNN algorithms within the domain of DL. The investigation unfolds across several key phases. Firstly, CNN algorithms in DL serve as the foundation for building a dynamic image target detection model. A spectrum of models is constructed, including a nine-layer convolution model, a seven-layer convolution model, and a residual module convolution model. Secondly, a simulation model catering to CNN-based image target detection is devised, grounded in optical imaging. Simulation experiments encompassing simple and salient environments, complex and salient environments, and intricate micro-environments are meticulously executed. Within a fixed training regimen, an extensive comparison is conducted, evaluating precision, accuracy, IoU, and FPS across different model states. Finally, this article introduces an attention mechanism within DL, paving the way for an attention mechanism CNN target detection model based on datasets. This model is then benchmarked against conventional target detection algorithm models. Subsequently, the accuracy and detection

time of these models in IoT object detection scenarios are thoroughly analyzed. The innovation of this article stems from the integration of the CNN algorithm within DL for dynamic image target detection. This article systematically examines the performance of diverse models across varied environments by constructing numerous model states and simulation models rooted in optical imaging. In addition, the incorporation of the attention mechanism within DL offers a more efficient and precise solution for image target detection within IoT systems. Beyond this, the article explores the application of DL and attention mechanisms within the IoT domain, providing valuable theoretical and practical insights for the future development and implementation of related technologies. This article is dedicated to the intricate realm of image object detection within IoT systems, with the primary objective of elevating detection performance by integrating DL and CNNs. The significance of this article is underscored by its direct relevance to practical applications. The key contribution of this article resides in its innovative approach, a fusion of CNN algorithms with DL frameworks, engineered to amplify image object detection capabilities within IoT systems. The study's framework encompasses the creation of multiple CNN models, including nine-layer convolution models, seven-layer convolution models, and residual module convolution models, all subjected to comprehensive performance comparison experiments across diverse environmental settings. Furthermore, the article leverages simulation models rooted in optical imaging principles, designed to simulate a variety of environmental conditions more realistically, thus providing a robust evaluation of performance. Of particular note, this article introduces attention mechanisms within the domain of DL. It introduces a dataset-based attention mechanism CNN object detection model, which excels in elevating both detection accuracy and efficiency. However, this research extends beyond the mere realm of technology. It boldly applies these findings to practical scenarios within the IoT domain, imparting invaluable insights that will steer the development and implementation of future IoT systems. Consequently, these contributions work harmoniously to augment the practicality and performance of IoT systems, marking a significant stride toward realizing efficient and effective IoT applications.

The article's framework and structure are delineated as follows:

1. Introduction: This section serves as the article's inception, setting the stage by elucidating its background, objectives, and methodological approach. It emphasizes the novel contributions and innovations while underscoring the constraints of conventional object detection methods regarding feature extraction and articulating the potential and challenges presented by DL technology.

2. Literature Review: This segment delves into the constraints confronting traditional object detection methods in feature extraction, particularly within the domains of information technology and artificial intelligence. As artificial intelligence continues to advance, conventional feature extraction techniques evolve toward DL algorithms. The section elaborates on the critical role of convolutional neural networks in feature extraction, particularly in the context of object detection within IoT systems.

3. Materials and Methods: Here, readers are acquainted with key concepts related to image object detection algorithms, IoT systems, computer vision, and image object detection theory. It elucidates the core tenets and functions of DL and convolutional neural

networks. Additionally, this section expounds the experimental design, encompassing scope, methodologies, and experimental configurations, providing clarity on the research's procedural facets, including the conduct of experiments and data collection.

4. Summary and Discussion of Results Analysis: This part ushers in the research's denouement, presenting the outcomes of simulation experiments across simple and complex environments, along with complex microenvironments. A comprehensive analysis of the detection outcomes of the CNN algorithm model, incorporating an attention mechanism, offers insights into the model's performance under varying environmental conditions. A meticulous examination of the efficacy of innovative and optimized techniques is presented in conjunction with the delineation of potential limitations and suggestions for future research directions.

5. Discussion: This section aligns with the research's fundamental objectives, engaging in an in-depth discussion of the research outcomes, particularly focusing on performance comparisons across diverse models within varying environments. It accentuates the effectiveness of innovative and optimized methods while hinting at potential limitations and charting the course for future research directions.

6. Conclusion: In this concluding segment, the research findings and contributions are succinctly encapsulated, accentuating their practical application potential and broader significance. Moreover, it extends recommendations and guidelines for prospective research endeavors.

## LITERATURE REVIEW

In the current landscape of information technology and AI technology, traditional target detection methods are grappling with limitations in feature extraction (*Tian et al., 2023*). As AI advances, a paradigm shift from conventional feature extraction techniques to DL algorithms, particularly CNNs, has emerged. Utilizing CNNs for feature extraction has led to heightened efficiency and accuracy in detecting targets within IoT systems (*Li et al., 2023a*). Notably, *Li et al. (2023a)* and *Li et al. (2023b)* demonstrated that a multi-scale analysis modulation recognition network employing denoising encoders, deep adaptive threshold learning, and multi-scale feature fusion significantly improved recognition accuracy in low signal-to-noise ratio environments. Meanwhile, the network shows the adaptive learning ability of different noise thresholds and the advantages of effective feature fusion modules under various modulation types (*Li et al., 2023b*). *Liao & Liu (2023)* proposed a CNN based on a depth-invariant network, which effectively boosts data detection efficiency and accuracy in image target detection. *Zhong et al. (2023)* advocated the fusion of real-time monocular 3D detection networks and CNNs to overcome temporal dependencies, resulting in improved accuracy for target detection. *Ding & Li (2023)* highlighted the advantages of DL-based target recognition, who demonstrateed superior performance compared to traditional AI algorithms. Their findings underscore DL technology's potential for enhancing image recognition. *Idris, Ya'u & Ali (2023)* showcased how CNNs coupled with non-maximum suppression techniques enhance context and deep feature extraction. This contributes to effective road crack detection by mitigating target

occlusion. *Park (2023)* explored CNNs integrated with multimodal information, extracting features from comments and images to accurately predict customers' revisiting behavior. *Lindenheim-Locher et al. (2023)* focused on real-time multimodal 3D CNNs, emphasizing their pivotal role in processing higher-resolution images and consequently enhancing 3D image detection capacities. In accordance with prior research, it becomes readily apparent that DL algorithms have yielded remarkable achievements within the realm of IoT image object detection. Specifically, CNNs have played a pivotal role in advancing the precision and efficiency of image object detection within IoT systems. Several investigations have underscored the transformative potential of novel techniques, such as multi-scale analysis and attention mechanisms, in profoundly augmenting the performance of image object detection, particularly in environments characterized by low signal-to-noise ratios. Moreover, the pervasive applicability of DL technology spans a multitude of domains, encompassing applications as diverse as road crack detection, image recognition, and intelligent surveillance. Nevertheless, despite the considerable successes associated with the application of DL technology, certain limitations persist. Conventional object detection methods grapple with constraints in feature extraction, rendering them less efficacious in the intricate milieu of IoT. Furthermore, DL technology encounters bottlenecks in specific scenarios, including computational complexity and model selection-related issues. Furthermore, the existing research landscape exhibits relatively narrow coverage, marked by a dearth of comprehensive comparisons and evaluations encompassing different models and technologies. This article makes a substantial contribution in mitigating these limitations. Harnessing CNN algorithms nested within a DL framework has facilitated the development of multiple dynamic image object detection models, adept at accommodating the exigencies of diverse environmental conditions. Additionally, the formulation of simulation models based on optical imaging principles furnishes a comprehensive evaluation of the model's efficacy across a spectrum of environments, encompassing both simple and intricate scenarios. The integration of attention mechanisms confers a marked enhancement in the accuracy and efficiency of image object detection, thereby bestowing robust support for the application and advancement of IoT technology. This article holds the promise to propel the progression of image object detection, illuminating a pathway towards further development within the field.

## MATERIALS AND METHODS

### Image target detection algorithm

(1) IoT system

The IoT represents a versatile technology capable of real-time data collection, encompassing a wide array of parameters including sound, light, heat, electricity, chemical composition, mechanical properties, biological metrics, and geographical location (*Lv & Singh, 2021*). The IoT system is underpinned by four key technologies, which are systematically classified and depicted in Fig. 1.

IoT systems find extensive application across various industries and domains, encompassing manufacturing, agriculture, transportation, energy management, healthcare,

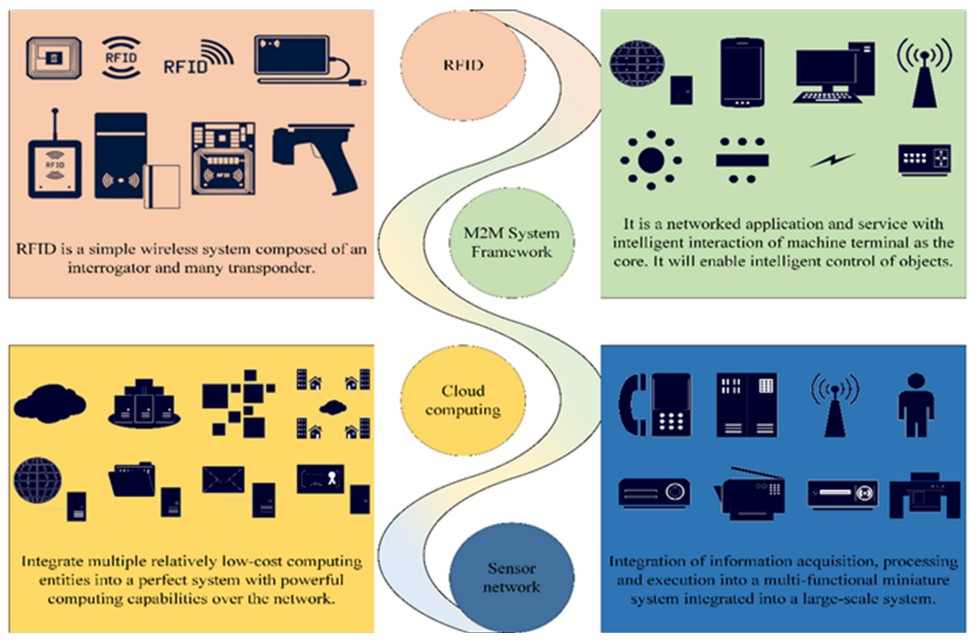

**Figure 1** **Key technologies of the IoT system.**

and more. In these contexts, IoT systems assume a pivotal role in facilitating intelligent development and refining resource allocation. They offer the mechanisms to boost operational efficiency, fine-tune resource utilization, and ultimately elevate the quality of life and working environments.

(2) Computer vision

Computer vision (CV) is a theoretical framework for processing visual information. This theory delineates the visual perception process into a multi-level, bottom-up analysis sequence. In this process, a series of distinct representations furnishes detailed information concerning the relevant visual environment (*Lv et al., 2021*).

Two challenges confront CV: the first pertains to feature extraction, and the second concerns the handling of vast computational datasets. DL offers a promising approach to addressing these challenges. Presently, CV encompasses eight distinct task categories, shown in Fig. 2.

As a pivotal domain in the AI field, CV exhibits extensive applications across various sectors, including industry, healthcare, security, agriculture, retail, human–computer interaction, culture, and the arts. Through the simulation of the human visual system, CV empowers computers to recognize, analyze, and comprehend image and video data, thereby enabling automatic image processing and analysis. The contributions of CV resonate in the form of innovation, efficiency, and convenience across diverse industries, propelling technological advancements and nurturing societal development, underscoring its undeniable significance.

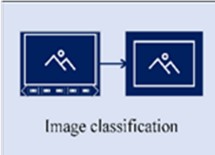 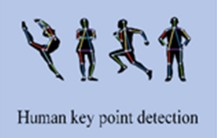 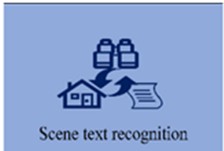 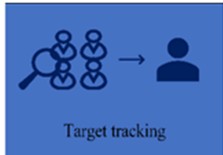

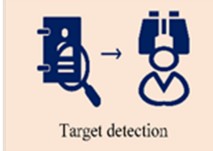 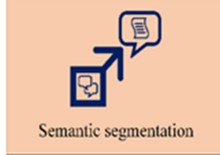 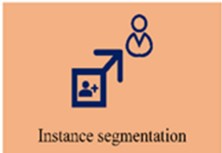 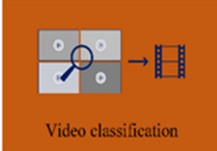

**Figure 2** Computer vision task classification.

(3) Image target detection

One of the primary challenges within the realm of CV is image object detection. The fundamental goal is to discern and categorize all objects of interest in an image while determining their respective positions. Object recognition has traditionally represented the most formidable hurdle in CV, primarily due to the inherent variability in appearances, shapes, and orientations of different objects, further compounded by factors such as varying illumination, occlusion, and other imaging-related complexities. Figure 3 illustrates the principal quandaries associated with image target detection.

There exist five conventional image object detection techniques, including edge detection, Hough Transform, binarization, projection histogram, and feature extraction. DL-based image target detection algorithms can primarily be divided into two categories: Two Stage and One Stage (*Zhao et al., 2019*).

## Deep learning (DL)

(1) DL

DL finds extensive applications in search technology, machine learning, data mining, machine translation, natural language processing (NLP), multimedia learning, recommendation systems, and personalized technology (*Janiesch, Zschech & Heinrich, 2021*; *Sarker, 2021*). Furthermore, DL algorithms demonstrate utility in predicting the geometric structure of RNA molecules (*Townshend et al., 2021*), inferring and forecasting various outcomes (*Ranganathan, 2021*), and evaluating collaborative work (*Pandian, 2021*).

DL research methods can be divided into three categories: neural network systems based on convolution, self-coding neural networks based on multilayer neurons, and deep belief networks with optimized neural network weights. Figure 4 displays the DL model.

(2) CNN

CNNs represent a class of algorithms within DL and also function as a type of artificial neural network, encompassing a series of mathematical operations and connection schemes (*Li et al., 2021*). CNNs are engineered to emulate the visual perception mechanisms observed in biological organisms (*Mamalakis, Barnes & Ebert-Uphoff, 2022*). They are

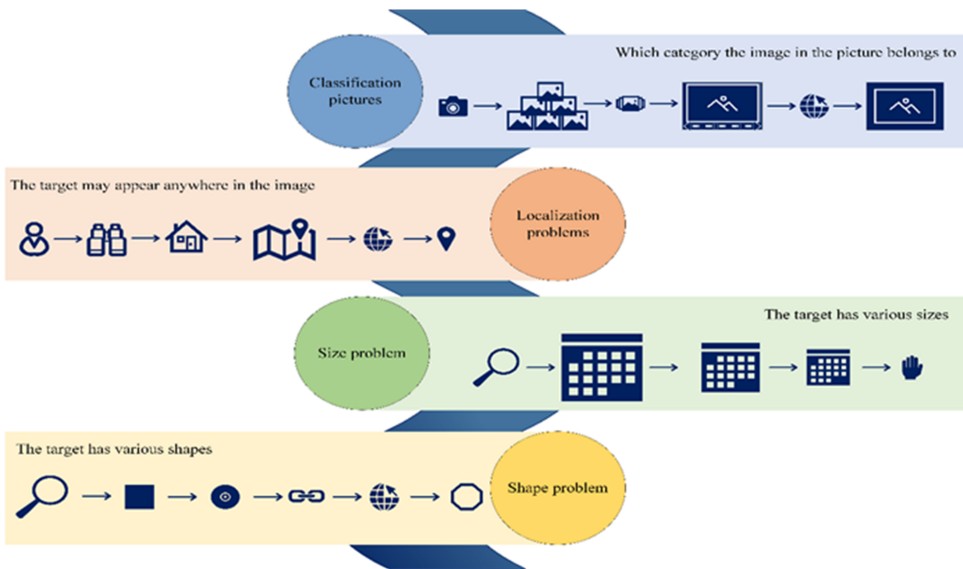

**Figure 3** **Key challenges in image target detection.**

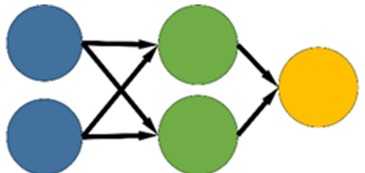

Feed forward neural networks

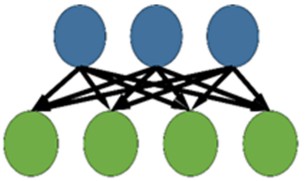

Restricted boltzmann machines

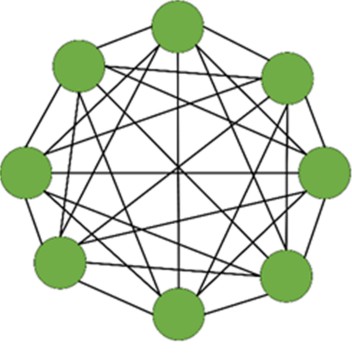

Hopfield network

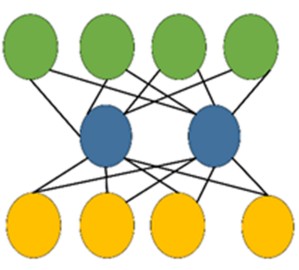

Autoencoders

**Figure 4** **DL model.**

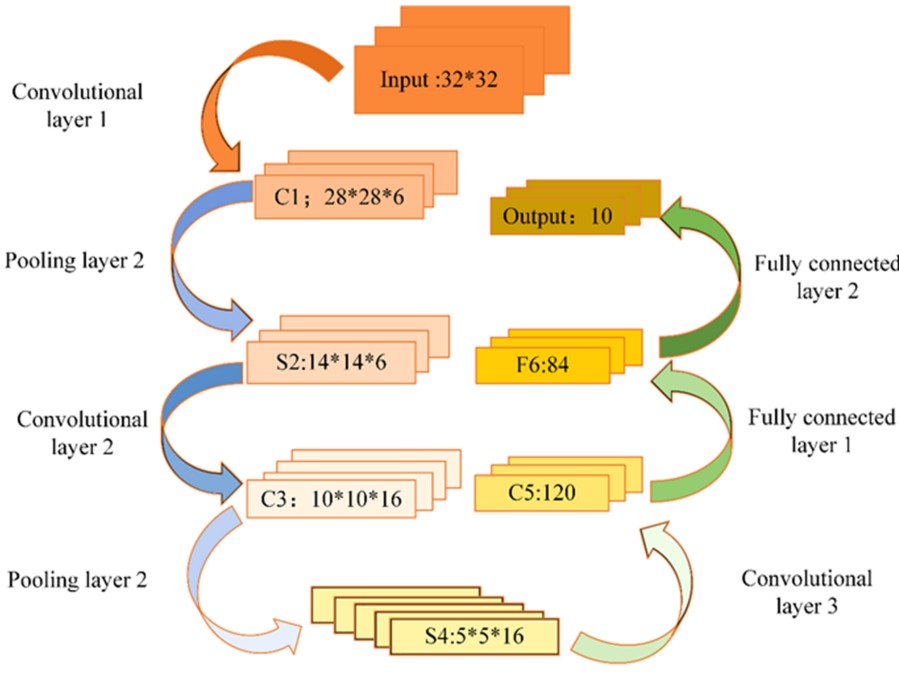

**Figure 5   CNN model.**

adaptable to both supervised and unsupervised learning and offer versatile applications across domains such as computer vision, natural language processing, agriculture (*Dhaka et al., 2021*), social sciences, healthcare (*Xu & Qiu, 2021*), and numerous others.

CNN architectures primarily comprise input layers, hidden layers, and output layers. Figure 5 shows the CNN model.

The input layer in CNN primarily serves as the data input interface and can effectively handle multidimensional data. Specifically, a one-dimensional CNN can process one- or two-dimensional (2D) datasets, whereas a 2D CNN can handle 2D or three-dimensional (3D) datasets. When dealing with four-dimensional datasets, a 3D CNN is a suitable choice. In the realm of CV, the 3D CNN is commonly employed.

Within the hidden layer of CNNs, one encounters pivotal components such as the convolutional layer, activation function, pooling layer, and fully connected layer. Among these, the convolutional layer occupies a prominent role. This layer comprises numerous convolution kernels, primarily tasked with the extraction of data features. The dimensions of the convolutional layer are contingent upon the size of the convolution kernel. Figure 6 illustrates a visual representation of convolution kernel operations.

The notation is defined as follows: $a$ represents the bias difference, $Q^L$ denotes the convolution input, $Q^{L+1}$ signifies the convolution output, $Q(x, y)$ represents the pixel of the feature map, $K$ stands for the number of channels of the feature map, $F$ indicates the size of the convolution kernel, $W_2$ denotes the convolution step, $O_K^{L+1}$ signifies the output size, and $P$ represents the number of filling layers. With these definitions in place, the sum of the input features in the receptive field can be expressed as follows:

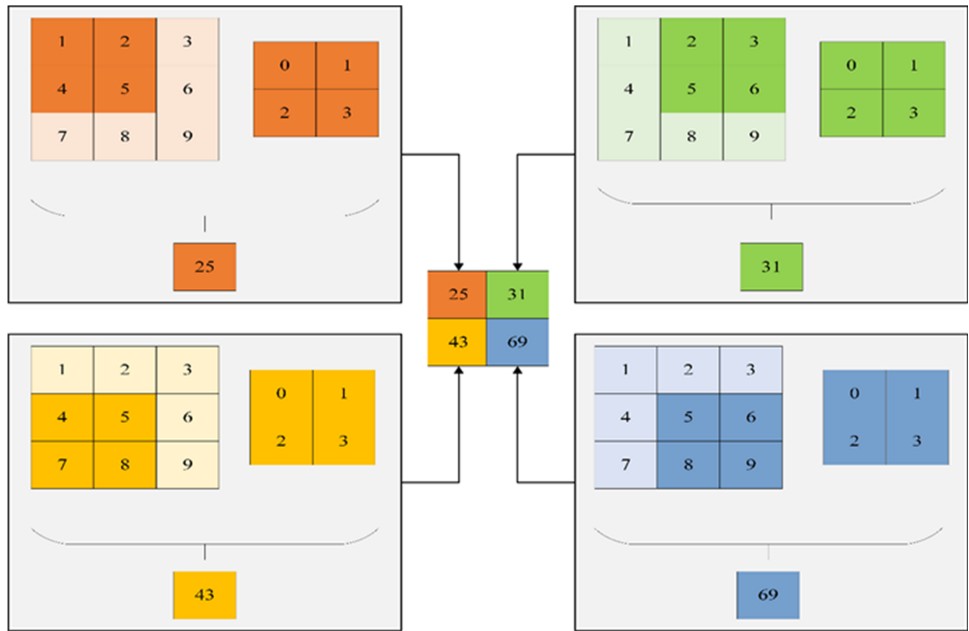

**Figure 6  Convolution kernel operation.**

$$Q^{L+1}(x,y) = \sum_{K=1}^{K_1}\sum_{X=1}^{F}\sum_{Y=1}^{F}\left[Q_K^L(W_1+x, W_2+y)O_K^{L+1}(x,y)\right]+a \tag{1}$$

$$(x,y)\in\{0,1,2,\ldots,n\}\, L_{n+1} = \frac{L_n+2P-F}{W_1}+1 \tag{2}$$

Activation functions can help express complex features within CNN. Several common activation functions include the rectified linear unit (ReLU), ReLU with slope, parametric ReLU, randomized ReLU, exponential linear unit (ELU), sigmoid function, and hyperbolic tangent function.

For instance, the ReLU function is calculated as Eq. (3), where $max(0,x)$ represents the ramp function in algebra.

$$F(x) = max(0,x) = \begin{cases} x & x \geq 0 \\ 0 & x < 0 \end{cases} \tag{3}$$

The sigmoid function is defined as Eq. (4), where $e$ represents a natural function and $y$ is a variable, with a value range between 0 and 1.

$$F(y) = \frac{1}{1+e^{-y}}. \tag{4}$$

The hyperbolic tangent function is given by Eq. (5), where $e$ represents the natural function and $x$ is the variable.

$$\tanh x = \frac{\sinh x}{\cosh x} = \frac{e^x - e^{-x}}{e^x + e^{-x}} \tag{5}$$

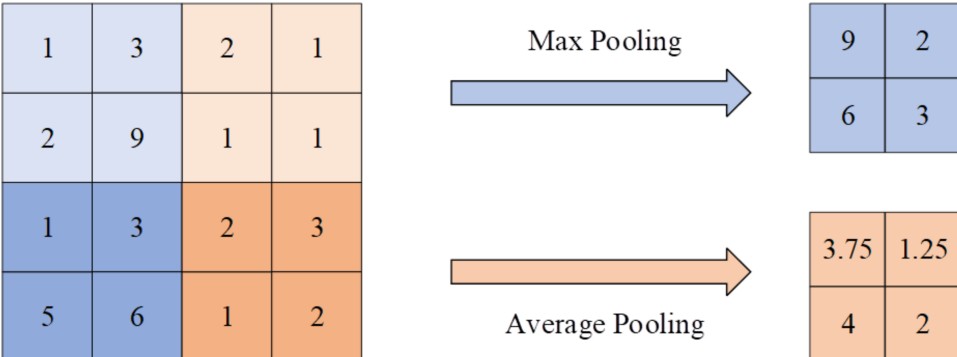

**Figure 7** Calculation process of pooling layer.

The pooling layer within the hidden layer plays a crucial role in feature selection and information filtering. The size of the pooling layer area is determined by the pooling layer size, the step size, and the number of filling layers. The output layer of the CNN is responsible for connecting the hidden layer and outputting the model's results. Figure 7 presents the calculation process of the pooling layer.

The pool layer is a pivotal component in CNNs, which is used to diminish the size of the feature map and extract essential features. The two most prevalent pooling operations are maximum pooling and average pooling.

Maximum pooling achieves downsampling by selecting the maximum value within a local area, thus retaining the primary features. The maximum pooling is given by Eq. (6).

$$OP(\alpha, \beta, \gamma) = MaxIP(\alpha s + i\beta + k\gamma) \tag{6}$$

In Eq. (6), $OP(\alpha, \beta, \gamma)$ represents the element of the pooled output feature map. $IP(\alpha s + i\beta + k\gamma)$ corresponds to an element of the input feature map. $(i, k) \in win$, signifying the coordinates within the window. $win$ denotes the window, and $s$ signifies the stride.

Average pooling accomplishes downsampling by calculating the average value within a local area, thus encapsulating the overall feature distribution. The average pooling is aarticulated as Eq. (7).

$$OP(\alpha, \beta, \gamma) = \frac{1}{winS} \sum_{(i,k) \in win} IP(\alpha s + i\beta + k\gamma) \tag{7}$$

Here, $winS$ signifies the number of elements in the window.

By employing both maximum pooling and average pooling, the pooling layer can reduce the size of the feature map, enhance the calculational efficiency, and bolster the robustness and generalization capabilities of the model while retaining crucial features. These operations are widely used in various layers of CNN, facilitating the effective processing of images and other data types.

To fine-tune the CNN target, the utilization of a loss function proves instrumental in predicting weights through coordinate adjustments. Among the prominent loss functions are loss functions are the mean squared error (MSE) loss function, cross entropy, Euclidean

distance loss function, Manhattan distance loss function, smooth L1 loss function, and Huber loss function.

MSE serves a frequently employed loss function, assessing the square of the average difference between the predicted value and the actual value. The MSE is calculated using Eq. (8).

$$MSE = \frac{1}{Z}\sum_{X=1}^{Z}\left(L^X - \ddot{L}^X\right)^2 \tag{8}$$

In Eq. (8), $Z$ represents the number of samples, $L^X$ denotes the real value of the $X$ sample, and $\ddot{L}^X$ signifies the predicted value of the $X$ sample.

The image target detection algorithm based on the neural networks can employ the MSE loss function, and its calculation equation is as follows:

$$Loss = \sum_{y=0}^{x^2} CoordError + IoUError + ClassError \tag{9}$$

In Eq. (9), $Loss$ represents the loss function; $x$ and $y$ are variables; $IoUError$ denotes the confidence value error; $ClassError$ signifies the classification error; $CoordError$ refers to the coordinate error.

If $x$ and $y$ represent two probabilities, the probability's cross entropy equation is defined as:

$$C(x,y) = -\sum_{n} x(n)\log y(n) \tag{10}$$

The normalized exponential function is adopted to optimize the cross entropy, and the following equation is derived:

$$Softmax(x) = \frac{e^{xn}}{\sum_{n} e^{xn}} \tag{11}$$

## Experimental design

(1) A training model for dynamic image target detection built based on the CNN algorithm

This article conducts all the experiments related to building a training model for dynamic image target detection utilizing the CNN algorithm on the Titan algorithm platform. The training of our model is performed using the Jetson TX2 embedded algorithm, with the Pascal VOC 2012 dataset used as the background database to augment the database. Several key configurations are established for the training process, including:

(1) Definition of the training database location for image targets and background data.

(2) Specification of classification labels.

(3) Determination of image dimensions during CNN training.

(4) Selection of the number of samples for each training session.

(5) Assignment of image target categories.

(6) Setting precise values for the image target detection algorithm.

(7) Configuration of the number and size of anchor boxes.

(8) Utilization of three distinct network models: a nine-layer CNN model, a seven-layer CNN model, and a CNN model with residual modules.

(9) Establishment of a target loss function.

(10) Utilization of the Adam optimizer for training algorithm optimization.

The Pascal VOC 2012 dataset is a versatile resource suitable for tasks such as image classification, object detection, and image segmentation. This dataset comprises an extensive collection of 23,080 image datasets and 54,900 project datasets, making it a benchmark dataset for image target detection. This article curates a training set consisting of 3,500 images from this dataset.

(2) CNN simulation models based on CV

This article develops several CNN simulation models grounded in computer vision principles. These models vary in architecture, and their specifications are detailed as follows:

(1) For the nine-layer CNN model, the number of network convolution layer channels follows the progression of 64, 128, 256, and 512, with an output set at $13 \times 13 \times 5 \times 7$.

(2) The seven-layer CNN model has an input layer size of $512 \times 512 \times 35$, with the number of convolution kernels specified as 8, 16, 32, 64, and 128.

(3) In the residual module CNN model, the size of the convolution layer is established at $3 \times 3 \times 140$.

(4) In these CNN simulation models, designed within the realm of visual computing, key parameters are set as follows:

(5) Initial learning rate: 0.0001

(6) Optimization method: Adam optimizer

(7) Number of iterations: 10

(8) Learning rate decay: 0.1

(9) Training cessation criterion: When the loss function remains unchanged for three consecutive weeks.

Furthermore, the architecture details are as follows:

(1) The seven-layer CNN model comprises seven layers, each with 16 neurons.

(2) The nine-layer CNN model incorporates nine layers, with each layer containing 32 neurons.

(3) The residual module CNN model consists of seven layers, with each layer featuring eight neurons.

In the evaluation process, this article conducts multiple training sessions to determine the impact of varying network models and loss functions on precision, accuracy, IoU, and FPS. Here, the article clarifies the definitions and significance of these performance metrics:

(1) Precision quantifies the proportion of correctly identified positive samples out of all samples that the model predicts as positive. It serves as an essential measure of the model's ability to avoid false positives.

(2) Accuracy represents the ratio of correctly predicted samples to the total number of samples across all categories. This metric provides a holistic assessment of the model's overall performance.

(3) IoU is not a commonly used metric in target detection. Its purpose is to gauge the degree of overlap between the model's detected target area and the actual target area. The IoU is calculated by dividing the intersection area of the target frame by the union area of the target frame. Higher IoU values indicate that the model's detection aligns closely with the actual target.

(4) FPS signifies the number of frames the model can process in one second when analyzing images or videos. It serves as a critical metric for assessing the model's real-time processing capability and performance in tasks requiring quick responses.

These performance indicators collectively provide a comprehensive evaluation of the model's effectiveness in the domain of target detection. It's essential to note that each metric emphasizes different aspects of performance, contributing to a comprehensive understanding of the model's capabilities and effectiveness in specific use cases.

This article is conducted on the Titan algorithm platform, with testing executed using the Jetson TX2 embedded algorithm. The article employs the Pascal VOC 2012 dateset as the background database, with a selection of 3,000 pictures images serving as the model's test set. TensorFlow framework was employed to implement the sample code for the target detection model. The model architecture encompasses a nine-layer CNN, a seven-layer CNN, and a residual module CNN model. The main function of this code is to define the CNN model, along with its corresponding loss function and optimizer. Throughout the training process, the loss function and accuracy metrics are computed at the conclusion of each epoch.

Based on prior studies, this article sets two labels for the binary classification of images, with the image dimension set at $512 \times 512 \times 3$. The selection of the object detection algorithm's accuracy and the configuration of anchor frames parameters draw from established practices in the field. In addition, the utilization of the Adam optimizer, as a commonly used training algorithm, is motivated by its demonstrated capacity to facilitate improved convergence and performance across various scenarios. The rationale for selecting the Adam optimizer is rooted in its extensive utilization and effectiveness with gradient descent optimization techniques (*Freitas et al., 2019*).

(3) Target image detection model of the CNN algorithm based on the attention mechanism.

The attention mechanism is prevalent in DL, enhancing the attention and processing ability of neural networks to input data. By automatically assigning varying weights to distinct segments of the input data, attention mechanisms prioritize crucial information during processing, thereby augmenting the model's performance and generalization capability. The structural layout of the target image detection model, employing the CNN algorithm integrated with an attention mechanism, is shown in Fig. 8.

This article evaluates the CNN-based target image detection model with an integrated attention mechanism using the KITTI dataset. A total of 3,600 images are designated for the training set, while 3,500 images constitute the test set. To ensure consistency, all images are standardized to a high resolution of $384 \times 1,280$. The model's performance is assessed based on the average accuracy of aerial view (AP-R40), which serves as a key performance index. Three levels of recognition difficulty—simple, intermediate, and challenging—are established, and the accuracy and detection time of various algorithmic

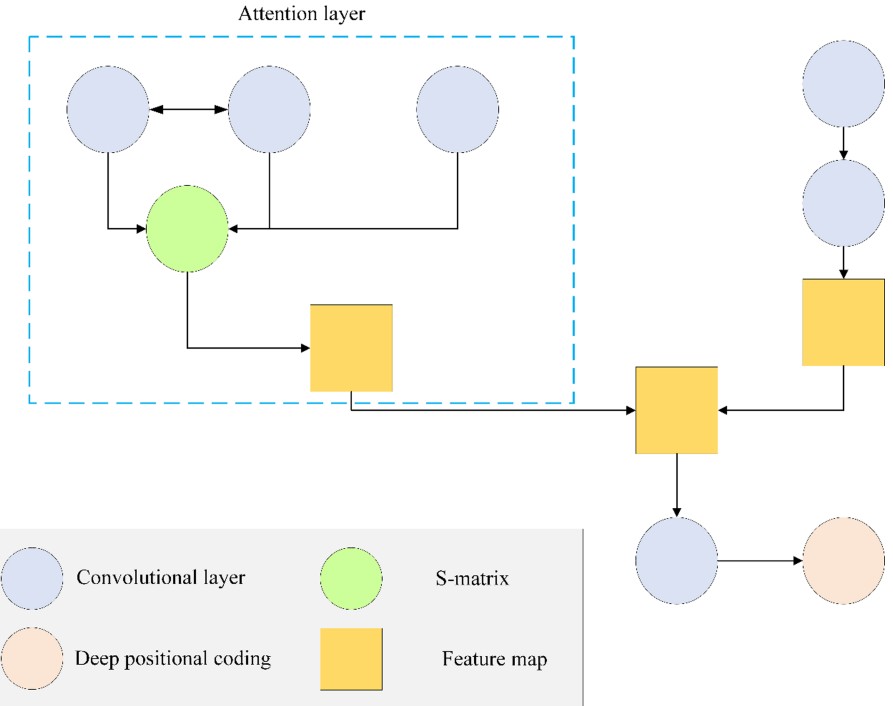

**Figure 8** **Structure of target image detection model of CNN algorithm based on attention mechanism.** Attention mechanism is a widely used technology in DL, which is used to enhance the attention and processing ability of neural network to input data. Attention mechanism automatically assigns different weights to different parts of input data, so as to pay more attention to important information in the process of processing, thus improving the performance and generalization ability of the model. The target image detection model structure of CNN algorithm based on attention mechanism is shown in in this figure.

models are compared across these conditions. Among these metrics, target detection accuracy reflects the algorithm's ability to correctly identify target objects and can be quantified by such indicators as precision, accuracy, IoU, and FPS. High accuracy suggests that the algorithm excels in precise target identification and localization, minimizing instances of misjudgment and missed detection, thus yielding more reliable detection outcomes. Detection time refers to the duration required for the algorithm to complete the target detection task, with short detection times ensuring swift responsiveness and adaptability to dynamic environments.

The KITTI dataset stands as an open dataset extensively employed in the fields of CV and autonomous driving. It comprises a diverse range of data types, including images, laser point clouds, calibration details, and vehicle trajectories, encompassing various scenes. This dataset is collaboratively developed through the Karlsruhe Institute of Technology in Germany and the Toyota European Research Center. It spans multiple different driving environments, such as urban streets and highways. This expansive dataset offers researchers a wealth of real-world data for studying and developing algorithms and technologies within the realms of target detection, object tracking, stereo vision, and autonomous driving.

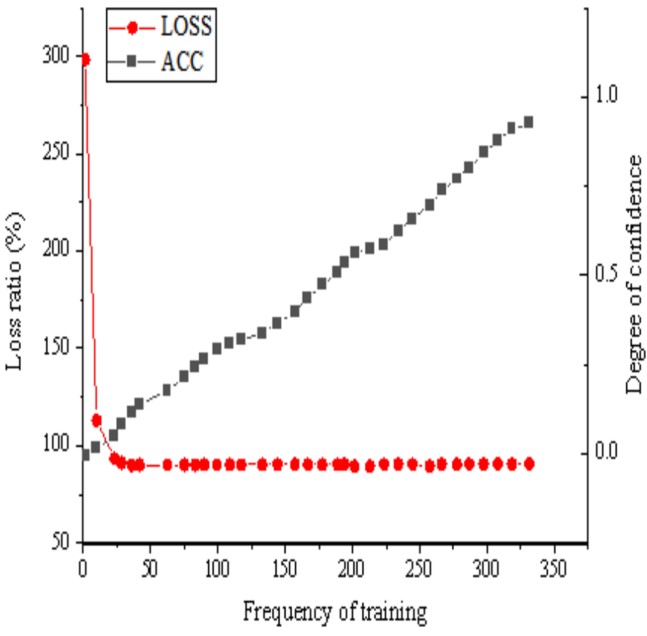

**Figure 9  Simulation results of the seven-layer CNN model in the simple and significant environment.**

This article conducts all experiments for constructing the target image detection model using the CNN algorithm based on the attention mechanism with the PyTorch framework. The ResNet-50 serves as the backbone network to acquire essential visual features related to the target. Additionally, the attention mechanism is used to reduce superfluous information in the image. Notably, the CNN algorithm model is configured with a batch size of eight, an initial learning rate of 0.0002, a learning attenuation rate of 0.1, a decay period of 50, and a weight attenuation coefficient of 0.0001. Optimization is achieved through the Adam optimizer, encompassing a total of 200 training rounds. The model consists of both coding and decoding blocks, each comprising six layers. The number of queries is set to 50, and the depth range spans from 0 to 60 m.

## RESULTS AND DISCUSSION

### Simulation of simple and salient environments

This article addresses the challenge of image target detection within simple and salient environments. To conduct simulation experiments, a seven-layer CNN model is used. The ensuing simulation results are shown in Fig. 9, showcasing the image target detection performance derived from the seven-layer CNN model within such simple and salient environments.

The analysis of Fig. 9 reveals that in a simple and salient environment, the utilization of a seven-layer CNN model appears to be generally suitable. Notably, the loss function within the image target detection algorithm steadily decreases as the training period extends, reaching stability once the training period reaches 12 units. The confidence curve for specific categories exhibits significant fluctuations throughout the training process,

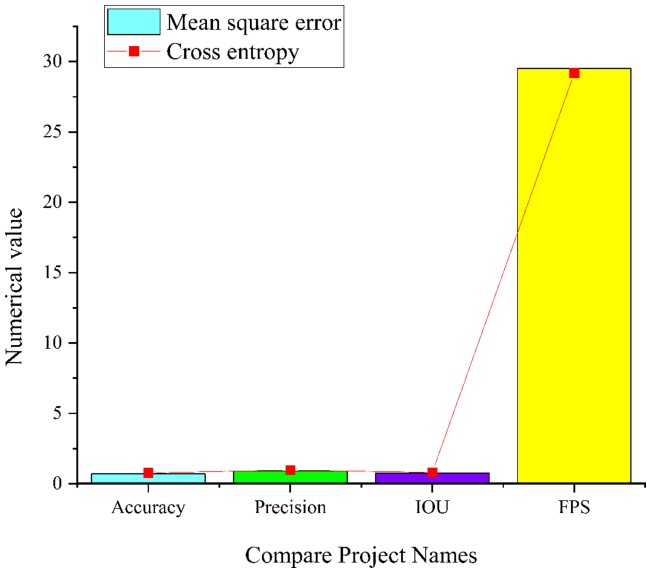

**Figure 10** **Simulation results of the nine-layer CNN model in the complex and significant environment.**

initially declining between 0 and 12 units. Subsequently, from 12: 00 to 89: 00, units into training, there is a discernible overall growth trend, albeit with minor fluctuations. Upon surpassing the 100-unit training threshold, the confidence curve for specific categories shows an upward trend, ultimately reaching a confidence level of 0.92 at the end of training. In this simple and salient environment, the seven-layer CNN model demonstrates high confidence and commendable accuracy. Nevertheless, some images remain undetected. This outcome might be attributed to the limited diversity of objects within the simple and salient environment, which potentially hinders the model's ability to sufficiently learn the characteristics of specific samples. In addition, factors such as model parameter configurations, data quality in the training dataset, and network structure may also affect the occurrence of undetected situations.

## Simulation of complex and salient environments

In the investigation of image target detection within a complex and salient environment, a nine-layer CNN model, a seven-layer CNN model, and a residual module CNN model are employed to conduct simulation experiments. The simulation outcomes are presented in Figs. 10–12, offering insights into the performance of image target detection based on the three CNN models in a complex and salient environment.

Figure 10 reveals that within the nine-layer CNN model in a complex and salient environment, the model employing cross entropy as the loss function achieves an accuracy of 0.77, a precision of 0.97, an IoU score of 0.81, and an FPS value of 29.2. Conversely, when the loss function is the mean square error (MSE), the model attains an accuracy of 0.7, a precision of 0.9, an IoU of 0.76, and a FPS value of 29.5. Comparing the two models, it becomes evident that the nine-layer CNN model employing cross entropy yields

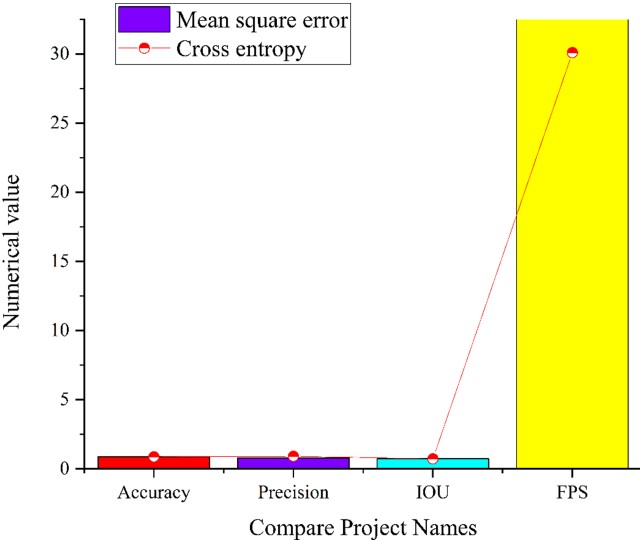

**Figure 11   Simulation results of the seven-layer CNN model in the complex and significant environment.**

higher precision, IoU, and accuracy in image target detection. However, it is noted that the detection speed is marginally lower compared to the nine-layer CNN model utilizing MSE. This observation aligns with the expected characteristics of cross entropy as the loss function in classification problems. While it enhances classification results, it may introduce a slightly complicated training process, thus affecting the detection speed.

Figure 11 provides insights into the performance of the seven-layer CNN model operating in a complex and salient environment. When employing cross entropy as the loss function, the model exhibits an accuracy of 0.87, a precision of 0.91, an IoU score of 0.73, and a FPS value of 30.1. Conversely, when utilizing MSE as the loss function, the model achieves an accuracy of 0.88, a precision of 0.79, an IoU value of 0.73, and a FPS value of 33.8. A comparative analysis between these two models reveals that the seven-layer CNN model using cross entropy as the loss function demonstrates superior accuracy in image target detection. However, it exhibits slightly lower detection speed and overall accuracy in comparison to the seven-layer CNN model utilizing MSE as the loss function. This phenomenon accords with the application of cross entropy in classification problems, which accentuates inter-class differences, ultimately enhancing accuracy. Nonetheless, it may have a minor impact on detection speed and overall model accuracy.

Figure 12 presents a comprehensive analysis of the performance of the residual module CNN model operating in a complex and salient environment under various conditions. When the number of training iterations is set to 1, the model using cross entropy as the loss function demonstrates an accuracy of 0.77, a precision of 1, an IoU score of 0.75, and an FPS value of 19.9. When employing MSE as the loss function, the model achieves an accuracy of 0.85, a precision of 0.98, an IoU of 0.8, and a FPS value of 20.4. Additionally, the model employing a combination of MSE and cross entropy as the loss function exhibits

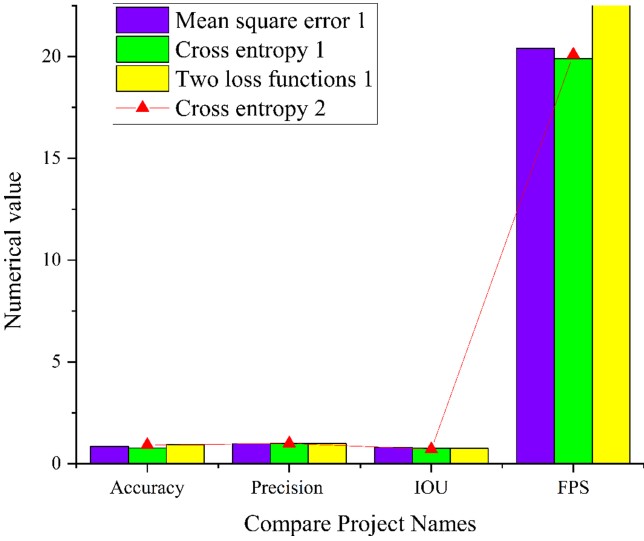

**Figure 12** **Simulation results of residual module CNN model in the complex and significant environment.**

an accuracy of 0.93, a precision of 0.99, an IoU of 0.76, and an FPS value of 26.3. With an increase in the number of training iterations to 2, the residual module CNN model with cross entropy as the loss function achieves an accuracy of 0.92, a precision of 1, an IoU of 0.71, and an FPS value of 20.1. Upon comparing the four models, it becomes evident that when the number of training iterations is set to 1, the residual module CNN model with loss function of MSE and cross entropy simultaneously attains higher precision, accuracy, IoU scores, and FPS values. These metrics were slightly lower than the CNN model using MSE as the loss function alone. The observed improvement in accuracy with increased training iterations is accompanied by a reduction in the IoU score. This phenomenon aligns with the iterative training process of DL models. The choice of different training strategies and loss functions can balance the model's performance between accuracy and IoU scores.

## Simulation of intricate micro-environments

In complex micro-environment, this article conducts comprehensive simulation experiments to evaluate the performance of the residual module CNN model within intricate micro-environments. The experiments are executed across three distinct network states, and the results are visualized in Fig. 13. It illustrates the model's capabilities for image target detection in various network states in intricate micro-environments.

The analysis of Fig. 13 sheds light on the performance of the residual module CNN model within an intricate micro-environment. When employing cross entropy as the loss function, the model achieves an accuracy of 0.85, a precision of 1, an IoU score of 0.8, and a FPS value of 28.4. Conversely, when utilizing MSE as the loss function, the model attains an accuracy of 0.97, a precision of 0.99, an IoU score of 0.75, and a FPS value of 28.6. The model employing both MSE and cross entropy as loss functions achieves an accuracy

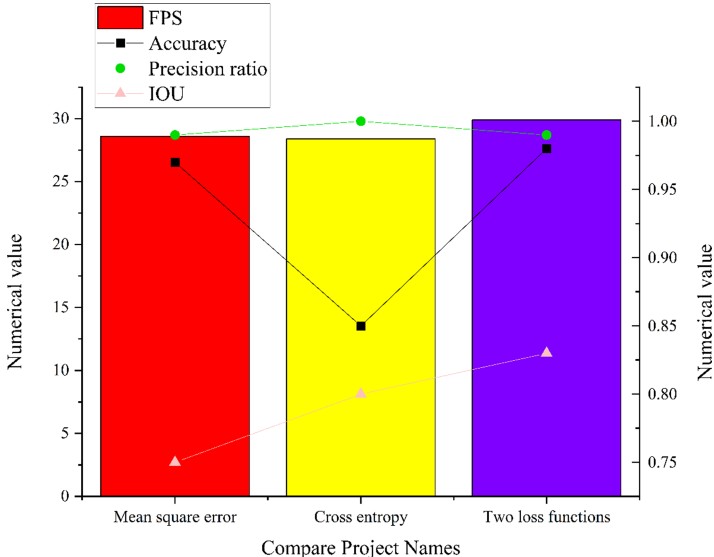

**Figure 13** **Simulation results of three different state models in the complex and micro environment.**

of 0.98, a precision of 0.99, an IoU score of 0.83, and an FPS value of 29.9. Comparative analysis reveals that the residual module CNN model employing a combination of MSE and cross entropy as loss functions excels in precision, accuracy, IoU scores, and FPS values. This result is attributed to the distinct effects that different loss functions impart on the model's optimization process, ultimately striking a balance between accuracy and other metrics.

## Analysis of target image detection results of the CNN algorithm model based on the attention mechanism

This article strategically employs a CNN model based on the attention mechanism for comparative experiments to explore the effect of this novel technology on image target detection within the IoT framework. Specific comparative results are presented in Fig. 14, which illustrates the juxtaposition of detection accuracy and detection time across various CNN algorithmic models and those integrating attention mechanisms in target image detection.

Upon scrutinizing Fig. 14, an examination of the accuracy comparison results for different algorithmic models reveals intriguing insights. It is apparent that the CNN model's detection accuracy, when exposed to multimodal information, is notably the lowest, registering at merely 14.5%. By contrast, CNN models employing non-maximum suppression or real-time monocular 3D detection networks exhibit relatively similar detection accuracies, standing at 19.72% and 20.1%, respectively. Remarkably, the CNN model operating within the depth-constant network framework demonstrates outstanding detection performance, boasting an impressive accuracy rate of 24.59%. Significantly, the CNN model reported here, fortified with an attention mechanism, yields substantial improvements in detection efficacy, showcasing a 0.27% accuracy enhancement compared

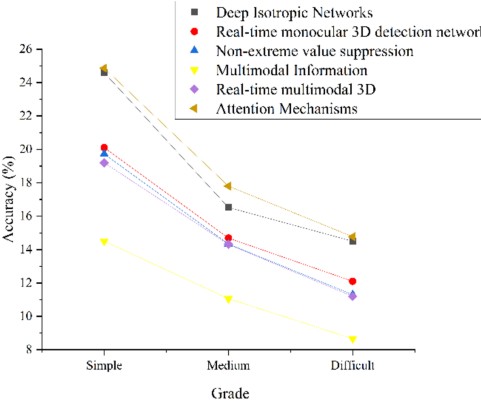

| Algorithm Name | Detection time(s) |
|---|---|
| Deep Isotropic Networks | 0.03 |
| Real-time monocular 3D detection network | 0.03 |
| Non-extreme value suppression | 0.03 |
| Multimodal Information | 0.03 |
| Real-time multimodal 3D | 0.03 |
| Attention Mechanisms | 0.05 |

(a) Comparison results of the accuracy of different algorithmic models

(b) Comparison results of detection time of different algorithmic models

**Figure 14** **Analysis and comparison results of target image detection results of different algorithm models.** According to the comparison results of the accuracy of different algorithm models, the figure shows that the detection accuracy of CNN under multimodal information is the lowest, only 14.5%. The detection of CNN under non-maximum suppression is similar to that under real-time monocular 3D detection network, with the accuracy of 19.72% and 20.1% respectively. The detection effect of CNN under the depth-constant network is excellent, and the accuracy is as high as 24.59%. The detection effect of CNN model based on attention mechanism used in this article has been significantly improved, and the accuracy has been improved by 0.27% compared with the CNN model under the depth-invariant network with the second performance. Under the medium and difficult levels, the accuracy ranking of the model has not changed, and the accuracy of the CNN model based on attention mechanism is 17.8% and 14.77% respectively. According to the comparison results of detection time of different algorithm models, it shows that the detection time of CNN based on depth-invariant network, CNN based on real-time monocular 3D detection network, CNN based on non-maximum suppression, CNN based on multimodal information and CNN based on real-time multimodal 3D are basically the same, all of which are 0.03s, and the detection time of CNN model based on attention mechanism is 0.05s. It is slightly higher than other models.

to the CNN model within the depth-invariant network, which claims the second-best performance. Under the medium and challenging difficulty levels, the accuracy ranking among models remains consistent, with the CNN model incorporating an attention mechanism, maintaining accuracy rates of 17.8% and 14.77%, respectively. Transitioning to the assessment of detection times of various algorithmic models, it becomes evident that CNN models predicated on depth-invariant networks, real-time monocular 3D detection networks, non-maximum suppression, multimodal information, and real-time multimodal 3D share nearly identical detection times of 0.03 s. Conversely, the CNN model based on an attention mechanism slightly extends its detection time to 0.05 s. In summary, the CNN model leveraging an attention mechanism exhibits superior detection performance and higher accuracy in the context of IoT target image analysis. However, as image complexity increases, the model's accuracy experiences a corresponding decline. Furthermore, a trade-off relationship exists between detection accuracy and processing time, where algorithms achieving higher accuracy invariably incur extended processing times.

## DISCUSSION

In this article, CNN algorithms are harnessed to tackle image recognition challenges within the realm of the IoT. A spectrum of dynamic image object detection models is crafted, encompassing diverse convolutional configurations, including nine-layer, seven-layer, and residual module convolutional models. These models facilitate an extensive exploration of performance across varying environmental complexities, spanning simple and significant settings, complex and significant contexts, as well as microenvironments. An attention mechanism is also introduced and seamlessly integrated into the CNN object detection models, permitting performance assessments at multiple levels. The findings demonstrate variations in computational complexity across distinct models. Particularly noteworthy, CNN models enhanced with the attention mechanism exhibited superior object detection accuracy while upholding lower computational complexity. This innovation accentuates the critical role of methodology, model states, and attention mechanisms in the pursuit of performance optimization. In contrast to prior research, this article introduces an innovative solution that combines the CNN algorithm with an attention mechanism for image object detection within the IoT domain. This integrated approach not only elevates the accuracy of image object detection but also effectively mitigates the need for manual identification, thereby offering a more intelligent and efficient method for dynamic image object detection within IoT systems. It fully leverages the potent capabilities of CNN in feature extraction. By constructing multiple dynamic image object detection models, this article significantly enhances the accuracy and efficiency of image object detection. The innovative contribution of this study resides in the proposal of a cutting-edge solution that maximizes the potential of DL technology and attention mechanisms to augment the accuracy and efficiency of image object detection within IoT systems. This article holds the potential to drive advancements in the field and promote the application and innovation of image object detection technology in IoT systems.

## CONCLUSION

This article strives to enhance the accuracy and speed of image object detection while fortifying the recognition capabilities of IoT systems. By leveraging the CNN algorithm in DL, a range of dynamic image recognition models were established, encompassing nine-layer, seven-layer, and residual module convolutional models. The article unveils substantial performance disparities among these models within varying environments. Notably, the residual CNN model showcases outstanding performance in complex micro-environments, achieving remarkable accuracy and efficiency. Furthermore, CNN models incorporating attention mechanisms demonstrate higher accuracy in IoT object image detection, opening up new possibilities for the field. This article heralds a new direction for optimizing and innovating image object detection technology. This article delves into the intricacies of image object detection and optimizes it within the framework of DL in the context of the IoT. This article bears substantial industrial significance, as the optimized model significantly amplifies the accuracy and efficiency of object detection. Consequently, this forms a robust foundation for automation and intelligence in various domains, such

as smart surveillance, autonomous driving, and intelligent manufacturing. Moreover, the incorporation of attention mechanisms contributes to the automatic detection of objects, thus reducing manual identification costs and elevating operational efficiency. This advancement positively impacts the application and proliferation of IoT technology, propelling industrial innovation and augmenting overall production efficiency.

While this article has made significant strides in IoT image object detection, certain limitations still exist. Firstly, the research scope is relatively constrained, as it did not comprehensively encompass various types of CNN models and lacks a comprehensive evaluation of other potential models. Future research should diversify the models, delving into each model's trade-off between accuracy and transmission speed to enhance object detection performance more precisely. Secondly, despite the introduction of attention mechanisms to improve object detection accuracy and efficiency, further research and optimization of this mechanism are required to achieve better performance. Additionally, future research could consider integrating other DL technologies to enhance image object detection performance. In conclusion, this article provides pivotal innovations and foundations for IoT image object detection. However, many avenues for future development exist, including broader model exploration, further optimization of attention mechanisms, and increased integration of DL technologies. Research in these areas will continue to propel the practical application and development of artificial intelligence image recognition technology in various domains.

### Funding
This work was supported by the Shaanxi Provincial Natural Science Basic Research Project under Grant No. 2022JM-374, the Open Projects Program of State Key Laboratory of Management and Control for Complex Systems under Grant No. 20220107. The funders had no role in study design, data collection and analysis, decision to publish, or preparation of the manuscript.

### Grant Disclosures
The following grant information was disclosed by the authors:
The Shaanxi Provincial Natural Science Basic Research Project: 2022JM-374.
The Open Projects Program of State Key Laboratory of Management and Control for Complex Systems: 20220107.

### Competing Interests
The authors declare there are no competing interests.

### Author Contributions
- Rui Chen conceived and designed the experiments, performed the experiments, performed the computation work, authored or reviewed drafts of the article, and approved the final draft.

- Lei Hei conceived and designed the experiments, performed the experiments, analyzed the data, prepared figures and/or tables, and approved the final draft.
- Yi Lai conceived and designed the experiments, performed the experiments, analyzed the data, performed the computation work, prepared figures and/or tables, and approved the final draft.

## Data Availability

The database used to build a dynamic image target detection training model based on convolutional neural network algorithm is available at PASCAL VOC 2012:

http://host.robots.ox.ac.uk/pascal/VOC/voc2012/index.html.

The code is available in the Supplemental File.

## Supplemental Information

Supplemental information for this article can be found online at http://dx.doi.org/10.7717/peerj-cs.1718#supplemental-information.

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
