# Peer review of "Object detection in optical imaging of the Internet of Things based on deep learning"

_PeerJ Computer Science, doi:10.7717/peerj-cs.1718_

## Round 0.1 · original submission · Major Revisions

1. The title of the paper needs to be further simplified to better reflect the main content of the paper.
2. The structure of the paper needs further improvement. Please refer to the relevant papers previously published in this journal.
3. There are too many grammar mistakes in the paper, please improve it.
4. The paper lacks some important references.

**Language Note:** The Academic Editor has identified that the English language must be improved. PeerJ can provide language editing services - please contact us at copyediting@peerj.com for pricing (be sure to provide your manuscript number and title). Alternatively, you should make your own arrangements to improve the language quality and provide details in your response letter. – PeerJ Staff

Reviewer 1 ·

Basic reporting

This paper is poorly organized which its structure is not logical enough for a decent research article. And, there are lots of grammar mistakes in the paper which makes it more ambiguous. From experimental results, the relevant results cannot support the hypotheses of the paper. By the way, the figures and tables are not included in the submitted manuscript.

Experimental design

From the manuscript, the models and implementation details are not clearly illustrated. Though the authors list three CNN networks of different purposes, the reviewer does not think they are meaningful. They are just some simple and common CNN networks that does not include new idea for improving accuracy and efficiency for object detection in IoT.

Validity of the findings

The novelty is limited. The paper just listed three different simple CNN network of stacked convolutional layers. Such designs are common in computer vision. The reviewer does not think these models benefit to object detection in terms of accuracy and efficiency in IoT. In another words, the authors fail to illustrate what the core novelty makes these simple CNN are suitable for IoT. Moreover, the results cannot support the conclusion.

Additional comments

N/A

Reviewer 2 ·

Basic reporting

1. Organization of the paper need to be improved.

2. Literature survey is missing and need to be modified based on current state of art methods. Some more paper based on current study in object detection.

3. Paper lacks novelty. Authors must show explain the novel contribution of the work with proper justification of the outcomes. What novelty is established in this work compared to existing works? Novel contribution of the work can be added at end of introductions with proper justification of the outcomes.

4. Explaining the problem and the gaps in existing literature in a concise but self-contained way (although readers might wish to consult references, they should not be forced to do so).

5. All figures need to be redrawn and its looks very poor. Resolution must be improved.

Experimental design

1. Specification of the implementation platform is missing.
2. How much data should be considered for training and testing for architecture implementation? Details of training and testing data sets must be tabulated.
3. To make the proposed algorithm of this article more readable use pseudo-code.
4. Layers details of proposed architecture must be included. Various Hyper parameters must be included.
5. . Various other performance metric need to be discussed and definitions of performance parameters must be included.

Validity of the findings

1. Comparative analysis of various performance parameters with respect to sate of art methods must be discussed. More recent state-of-the-art approaches should be compared; the experiments should use more sizable real-world data sets from public repositories (if any);
2. The computational complexity of the proposed work must be discussed. Also, compare the proposed method in terms of computational complexity?
3. Comparative analysis with respect to inference/fps and real-time time analysis is missing?
4. Visualized Results with respect to various categories of data sets must be discussed and presented.
5. Comparative analysis with respect to various other data sets like KITTI etc. is missing? The comparison can be a bit unfair if different data sets are not used for comparative analysis.
6. In result analysis, the author must explain the results in detail, with observations and general reasons. It is beneficial that the author includes the scientific basis for each finding.
7. Precision vs. recall curves of the proposed algorithms with respect to data sets must be included.
8. Authors must include results based on various occlusion conditions.

Additional comments

1. In all results tables’/figures utilized datasets like in table 1 etc. must be cited with proper and specific citations.
2. Future work and limitations of the proposed work can be added and discussed.
3. Add industrial significance of the proposed approach.
4. Has the Author implemented the architecture from scratch and identified the novel condition in deep networks.
5. Various Hyper parameters like learning rate, optimizer etc. need to be included.

Annotated reviews are not available for download in order to protect the identity of reviewers who chose to remain anonymous.

·

Basic reporting

The article proposes an image object detection model for the Internet of Things (IoT) that utilizes optical imaging visualization and deep learning. The study focuses on improving the accuracy and efficiency of image recognition in the IoT context. By employing a Convolutional Neural Network algorithm, the authors develop three different model states and conduct simulation experiments in various environments. The results show that the seven-layer CNN model performs well in simple and significant environments, with high confidence and accuracy. However, there are still some limitations. In complex and significant environments, the residual network model with the loss function of mean square error (MSE) and cross entropy proves to be the best option, achieving high precision, IOU, and accuracy. Similarly, in complex and micro environments, the residual CNN model with MSE and cross entropy achieves good results, with a precision of 0.99, IOU of 0.83, and FPS of 29.9. Overall, the residual network models with MSE and cross entropy exhibit strong performance in different environments, offering high detection accuracy. This work contributes to improving image object detection, enhancing IoT system recognition capabilities, and reducing manual recognition costs.
The article still has some shortcomings, as follows:
1) The abstract lacks a clear research objective. Is the purpose of this study to improve the accuracy and detection efficiency of image recognition technology in the IoT context using optical imaging visualization and deep learning techniques? This part needs to be clearly stated.
2) In the introduction, when describing the advantages of deep learning methods, it would enhance the credibility of the argument to cite more specific research findings and case studies related to image object detection in IoT systems.
3) What is the novelty of this article? Is it the integration of computer vision and CNN with image object detection algorithms in IoT systems?
4) In terms of IoT systems, in which fields and industries are they widely applied? What are some specific examples of intelligent development and resource allocation achieved through IoT systems?
5) What is the scope and importance of computer vision applications?
6) Regarding CNN, a more detailed explanation of the image pooling layer is needed, along with specific examples of pooling algorithms, to help readers better understand its function and purpose.

Experimental design

7) How were the classification labels and image dimensions chosen and set for the construction of the dynamic image object detection training model based on the CNN algorithm? Were these choices and settings based on previous research or experimental results? Is there a specific basis for determining the accuracy of the object detection algorithm, the number and size of anchor boxes? Additionally, was the Adam optimizer used for optimizing the training algorithm selected through experimentation, and why was it chosen?
8) The analysis in the results section is insufficient. For example, regarding Figure 8, a more detailed explanation is needed, including the trend of the loss function during training and the fluctuation of confidence coefficient curves for specific categories.

Validity of the findings

9) When describing the loss function, more specific definitions and examples need to be provided. For example, for the MSE loss function, the meaning of each term in the loss function can be explained, along with a specific calculation example, to help readers understand how the total loss is computed.

Additional comments

10) Why were only the nine-layer CNN model, seven-layer CNN model, and residual module CNN model compared when establishing the simulation model? Were other types of CNN models considered? Further research could explore the impact of other types of CNN models on CV-based image object detection algorithms in IoT systems.

---

## Round 0.2 · Minor Revisions

Please give an explanation of the contribution of the work with proper justification of the outcomes.

What new knowledge is established in your work compared to existing similar works?

Reviewer 2 ·

Basic reporting

Paper lacks novelty. Authors must show explain the novel contribution of the work with proper justification of the outcomes. What novelty is established in this work compared to existing works?

Experimental design

Paper lacks novelty. Authors must show explain the novel contribution of the work with proper justification of the outcomes. What novelty is established in this work compared to existing works?

Validity of the findings

Paper lacks novelty. Authors must show explain the novel contribution of the work with proper justification of the outcomes. What novelty is established in this work compared to existing works?

·

Basic reporting

The modified version has greatly improved. This paper aims to improve image recognition technology in the Internet of Things (IoT) by utilizing convolutional neural networks (CNN) and deep learning algorithms. Different model configurations were proposed and evaluated in various scenarios, showing promising results in terms of accuracy and detection efficiency. An attention mechanism was further incorporated into the CNN model, achieving even higher accuracy rates in IoT target image detection. The findings of this paper contribute to enhancing the effectiveness of object detection in IoT systems and provide a theoretical foundation for optimizing imaging and image target detection technologies.
But there are still some issues that need to be modified as follows:
1. The value of the research in this paper needs to be enhanced by a stronger discussion of the background and purpose of the study in the introduction section.
2. The introduction section also needs to focus on the specific framework and structure of the paper.
3. The last paragraph of the introduction section needs to highlight the discussion of the research contribution of this paper.
4. The literature review section needs to summarize the main results and shortcomings of the current research and, in doing so, point out the energy that the research in this paper provides to fill these shortcomings.

Experimental design

1. The methods section has more theoretical content, and these are the underlying consensus, so it is recommended that the theoretical content be cut and the modeling content be increased.
2. The structure of Figures 1-6 is too simple and needs to be increased in complexity and as a way to more clearly represent the research design of this paper.

Validity of the findings

1. The discussion section needs to highlight the strengths of this paper by discussing its findings and comparing them with more advanced research.
2. The conclusion section is overloaded and needs to be made concise and a discussion of future developments is needed. And the research limitations of this paper also need to be highlighted in the conclusion section.

---

## Round 0.3 · accepted · Accept

The author has completed all revisions; the manuscript can be published.